# Genome-Wide Characterization of the *Aux/IAA* Gene Family in Orchardgrass and a Functional Analysis of *DgIAA21* in Responding to Drought Stress

**DOI:** 10.3390/ijms242216184

**Published:** 2023-11-10

**Authors:** Miaoli Wang, Guanyan Feng, Zhongfu Yang, Jiahui Wu, Bingyan Liu, Xiaoheng Xu, Gang Nie, Linkai Huang, Xinquan Zhang

**Affiliations:** College of Grassland Science and Technology, Sichuan Agricultural University, Chengdu 611130, China; wangmiaoli2022@126.com (M.W.); feng0201@sicau.edu.cn (G.F.); yangzf211@163.com (Z.Y.);

**Keywords:** *Dactylis glomerata*, Aux/IAA gene family, gene expression, drought stress

## Abstract

Drought stress is an important factor that reduces plant biomass production and quality. As one of the most important economic forage grasses, orchardgrass (*Dactylis glomerata*) has high drought tolerance. Auxin/indole-3-acetic acid (*Aux/IAA*) is one of the early responsive gene families of auxin and plays a key role in the response to drought stress. However, the characteristics of the *Aux/IAA* gene family in orchardgrass and their potential function in responding to drought stress remain unclear. Here, 30 *Aux/IAA* members were identified in orchardgrass. Segmental duplication may be an important driving force in the evolution of the *Aux/IAA* gene family in orchardgrass. Some *Aux/IAA* genes were induced by IAA, drought, salt, and temperature stresses, implying that these genes may play important roles in responding to abiotic stresses. Heterologous expression in yeast revealed that *DgIAA21* can reduce drought tolerance. Similarly, the overexpression of *DgIAA21* also reduced drought tolerance in transgenic *Arabidopsis*, which was supported by lower total chlorophyll content and relative water content as well as higher relative electrolyte leakage and malondialdehyde content (MDA) than Col-0 plants under drought conditions. The results of this study provided valuable insight into the function of *DgIAAs* in response to drought stress, which can be further used to improve forage grass breeding programs.

## 1. Introduction

Orchardgrass (*Dactylis glomerata*) is one of the top four economically important forage grasses and is extensively cultivated; it plays a vital role in livestock husbandry and ecological remediation [1,2,3]. Compared with other perennial grass species, orchardgrass has more drought tolerance and shade tolerance. Auxin plays an important role in sensing and signaling related to varied abiotic stress conditions, such as drought, temperature, and salt stress [4]. Aux/IAA is a nuclear protein with a short life cycle that plays a key role in auxin balance in plants [5,6]. The *Aux/IAA* gene family contains four domains, including the LxLxLx leucine-rich repeat motif domain I, as well as a highly conserved sequence with GWPPV domain II, domain III, and domain IV, which are located at the C-terminus and bind to the auxin response factor [7,8]. Previous studies have identified the features of Aux/IAAs in plant species such as *Malus domestica* [9], *Hordeum vulgare* [10], *Hedychium coronarium* [11], *Zea mays* [12], *Brassica rapa* [13], and *Brassica napus* [14].

Drought is a widespread problem that negatively impacts the economy. At present, approximately 40% of the Earth’s total landmasses are suffering drought stress [15,16]. Drought stress in plants occurs if the evaporation and transpiration rate exceed the rate of water uptake by the roots [17,18]. Drought stress has adverse effects on seed germination, vegetative growth, flowering, and fruit ripening [19,20,21,22]. As sessile organisms, plant populations adapt to stresses in wild and agricultural habitats through changing their genetic and physiological–biochemical characteristics, as well as their morphological traits [23]. Closing the stomatal aperture to retain the leaf water potential is the first step in the response to drought stress [24]. A decrease in the entrance of CO_2_ into leaves due to stomatal closure can suppress the rate of photosynthesis. In addition, drought stress also creates ionic and osmotic imbalance, reactive oxygen species (ROS), and changes to cellular metabolism that could limit plant growth and development [21,25,26,27].

Increasing evidence has shown that *Aux/IAA* genes can respond to drought stress. Water loss by stomates was adjusted in *IAA12* and *IAA17* mutants through strongly clustering to respond water deficit stress [28,29,30]. Plants with mutations of *IAA5*, *IAA6*, and *IAA19* showed reduced drought resistance through downregulating aliphatic glucosinolate levels [31]. The overexpression of *OsIAA20* in rice increased the expression level of the abscisic acid (ABA)-responsive gene *OsRab21* in response to drought stress [32]. The overexpression of *MdIAA9* of tobacco had lower relative electrolyte leakage and MDA levels, as well as higher total chlorophyll content and proline content to enhance resistance to drought stress [33].

However, limited information is available about the *Aux/IAA* gene family in orchardgrass and the functional characteristics of *Aux/IAA* genes in response to drought stress. Therefore, in this study, a genome-wide characterization of the *Aux/IAA* gene family in orchardgrass was presented. Furthermore, this study analyzed the physiochemical information, cladogram, gene structure, conserved motif, chromosome location, and synteny of the *Aux/IAA* gene family in orchardgrass. The expression patterns of *Aux/IAAs* under different stress conditions were determined. Moreover, the overexpression of *DgIAA21* in Arabidopsis plants further verified the role of *DgIAA21* in drought stress. The results of this study provide insight into the mechanism through which the *Aux/IAA* gene family responds to abiotic stress conditions.

## 2. Results

### 2.1. Identification of the Aux/IAA Gene Family in Orchardgrass

Thirty Aux/IAA proteins were identified in orchardgrass; the protein sequences are shown in Appendix A. The lengths of DgAux/IAA proteins varied from 89 to 530 amino acids, with an isoelectric point ranging from 4.62 to 10.16 (Appendix A). The lowest molecular weight was 23.45 KDa, and the highest molecular weight was 56.59 KDa. Furthermore, the instability index of DgIAA16b, DgIAA24, and DgIAA26 was low, which showed that these three proteins were relatively stable. The grand average of hydropathicity of DgAux/IAA proteins was a negative value, except for DgIAA27, indicating that nearly all proteins were hydrophilic and unstable in water. The physiochemical analysis showed that the proteins of DgIAAs are acid, alkaline, or acid–base, most of which are unstable and hydrophilic. Subcellular localization indicated that DgAux/IAA proteins were mainly located in the nucleus and chloroplasts, implying that DgAux/IAA proteins may play an important role in the nucleus and chloroplasts.

### 2.2. Motif Analysis and Phylogenetic Relationships Reveal the Evolution of Aux/IAA Genes

To explore the evolution of the *DgAux/IAA* family, a phylogenetic tree was constructed, the structure of exons and introns was investigated, and conserved motifs were further analyzed (Figure 1). Based on the phylogenetic tree, DgAux/IAA proteins were separated into two clades: A and B. Clade A had higher proportion of DgAux/IAA protein members (60%). The number of introns varied from zero to seven. Almost all *DgAux/IAA* genes had an intron, except for *DgIAA9. DgIAA31a*, *DgIAA31b*, *DgIAA23*, *DgIAA18a*, and *DgIAA33* had one intron, while the other genes had two or more introns. Additionally, *Aux/IAA* genes in cluster A had at most four introns, but many genes in clade B had more than four introns, such as *DgIAA27*, *DgIAA6*, *DgIAA7*, and *DgIAA18b.* Motif 2 existed in all DgIAA proteins, and motifs 1, 3, and 4 existed in nearly all DgIAA proteins. Motif 5, motif 9, and motif 10 only existed in clade A, while motif 8 was only present in clade B, suggesting DgIAA proteins were highly conserved in the clades.

To investigate the phylogenetic relationships among the Aux/IAA proteins, a phylogenetic tree was constructed using 90 protein sequences, including 30 sequences from *Dactylis glomerata*, 29 sequences from *Arabidopsis thaliana*, and 31 sequences from *Oryza sativa* (Figure 2). All of the 90 Aux/IAA proteins were divided into two classes (A and B) based on the classification of the *Arabidopsis thaliana* Aux/IAA gene family in previous research [34]. The Aux/IAA proteins of orchardgrass were tightly clustered with rice but distant to Arabidopsis; for example, OsIAA21 and DgIAA21, AtIAA14 and AtIAA17 were found together. These results illustrated that divergence has occurred between dicots and monocots in *Aux/IAAs*.

### 2.3. Chromosome Distribution and Synteny Analysis of DgAux/IAAs

Figure 3 shows the results of the chromosome distribution and synteny analyses of *DgAux/IAAs*. *DgAux/IAA* genes were randomly located on six chromosomes, but no *DgIAAs* were found on Chromosome 3 (Figure 3A). Eight predicted genes were present on Chromosome 4, followed by six genes and five genes on Chromosomes 5 and 6, respectively. Four genes were found on Chromosomes 2 and 7, while two genes were distributed on Chromosome 1, which had lower gene density than the other chromosomes. Furthermore, we found that some *DgIAA* genes were adjacent to each other. For example, *DgIAA12* and *DgIAA30a*; *DgIAA15a* and *DgIAA16b*; and *DgIAA31a*, *DgIAA31b,* and *DgIAA30b* were connected in series, indicating that tandem duplication occurred in these *DgAux/IAAs.*

To uncover the origin and evolution of the *DgAux/IAA* members, a synteny analysis was performed in orchardgrass as well as among orchardgrass, Arabidopsis, and rice. Seven segmental duplication pairs of *DgIAAs* were found in orchardgrass, while Chromosomes 5 and 6 had the most duplication regions (Figure 3B). The number of collinear gene pairs between orchardgrass and rice was significantly higher than that between orchardgrass and Arabidopsis, revealing that orchardgrass *DgIAAs* showed more homology with rice than Arabidopsis, supporting a close relationship with monocotyledons during evolution (Figure 3C). *DgIAA4* and *DgIAA13* in orchardgrass had collinearity both with *Aux/IAA* genes in Arabidopsis and rice, suggesting that *DgIAA4* and *DgIAA13* may have existed before the differentiation of dicots and monocots.

### 2.4. Expression of DgIAA Genes in Different Tissues and Their Response to Different Stresses

The expression patterns of *DgIAA* genes were observed in the spike, stem, leaf, flower, and root tissues (Figure 4A). The expression of *DgAux/IAA* genes varied in different tissues, with the majority of *DgAux/IAA* genes highly expressed in the spike and root. For example, *DgIAA1*, *6*, *9*, *12*, *18a*, and *24* were specifically highly expressed in the root tissue. These results implied that *DgAux/IAA*s may be involved in both spike and root development.

Previous research has indicated that the *Aux/IAA* gene family responds to different stresses; therefore, we investigated the expression of *DgAux/IAA* gene family members under five stress conditions (Figure 4B). For IAA treatment, the expression level of the *DgIAA6* and *DgIAA21* genes maintained a significant decreasing trend, while the expression level of the *DgIAA16a* and *DgIAA33* genes maintained a significant increasing trend. The expression level of the *DgIAA4*, *16a*, and *21* genes maintained a significant decreasing trend under drought stress. For salt stress, the expression level of the *DgIAA3*, *4*, *6*, *16a*, *21*, and *33* genes exhibited oscillatory changes. But the expression level of the *DgIAA30a* gene sustained a significant decrease. For high-temperature stress, the expression level of the *DgIAA6* gene sustained a significant reduction, and the expression level of the *DgIAA33* gene sustained a significant increase. The expression level of the *DgIAA3*, *4*, *6*, *21*, *30a*, *and 33* genes maintained a significant decreasing trend under low-temperature stress.

### 2.5. Heterologous Expression of DgIAA3, DgIAA4, DgIAA6, and DgIAA14 Genes Enhances Drought Tolerance in Yeast

To further analyze the function of *DgIAAs* under drought stress, some genes were transformed into the pYES2 vector and then introduced into INVScI. *DgIAA3*-, *DgIAA4*-, *DgIAA6*-, and *DgIAA14*-pYES2 stains appeared to be highly insensitive due to higher cell recovery in 100- and 1000-fold dilution rates under a high sorbitol level (2.5 M), apart from *DgIAA21*-pYES2 stains. This indicates that *DgIAAs* may exhibit different patterns of behavior when subjected to drought stress conditions (Figure 5).

### 2.6. Overexpression of DgIAA21 Negatively Regulates Root Length under Drought Stress

The fluorescence signal of *DgIAA21*-GFP fusion vector was obtained primarily in the nucleus, and minority in nucleus and cytoplasm (Figure 6). To verify the gene function under drought stress, we selected 3 transgenic *DgIAA21* Arabidopsis lines for further analysis (Appendix A). The germination rate was lower in *DgIAA21*-overexpressing lines than Col-0 in 300 mM mannitol (Figure 7A). The root lengths of Col-0 plants increased significantly compared to OE plants in 300 mM mannitol concentration (Figure 7B).

### 2.7. Overexpression of DgIAA21 Negatively Regulated Drought Tolerance in Transgenic Arabidopsis Plants

Three-week-old *DgIAA21-*overexpressing and Col-0 plants were treated without water for 14 days and re-watered for 4 days, and the result showed that the *DgIAA21-*overexpressing plants had fewer green leaves than Col-0 plants (Figure 8A). Under control conditions, the photosynthesis parameters showed that transgenic plants had a greater intercellular CO_2_ concentration (Ci), stomatal conductance (Gs), and transpiration rate (Tr) than Col-0 plants, while the net photosynthetic rate (Pn) and water use efficiency (WUE) were lower than those in Col-0 plants (Figure 8B–F). Moreover, transgenic *DgIAA21* lines had higher water loss and lower total chlorophyll content than Col-0 plants (Figure 8G–I). Under drought conditions, the relative electrolyte leak in transgenic lines was higher, whereas the relative water content was lower than in Col-0 plants (Figure 8H,K). In addition, the transgenic plants, except for the OE3 lines, exhibited significantly higher MDA content, while transgenic plants had lower Fv/Fm and total chlorophyll content than those of Col-0 plants under drought stress conditions (Figure 8I,J,L). We measured eight chlorophyll-related genes to explore the impact on total chlorophyll content. Chlorophyll synthesis gene *CDR1* was significantly reduced under normal conditions (6.4-fold) and drought conditions (2.3-fold), suggesting that the overexpression of *DgIAA21* transgenic plants could decrease the total chlorophyll content through reducing the expression level of the *CDR1* gene (Figure 9B,C).

To determine the molecular mechanism of sensitive phenotypes of *DgIAA21*-overexpressing transgenic plants, 13 stress-related genes were analysis under drought stress (Figure 9A). *APX1* and *NHX2* gene expression levels were decreased in the transgenic line compared with the Col-0 type. *CAT1* and *SOD1* gene expression levels were steady within 24 h after treatment. *SOS1* (12h, 2.0-fold), *RD29B* (6 h, 2.2-fold), and *ABI1* (6 h, 2.0-fold) gene expression levels in the transgenic line were significantly decreased than those of the Col-0 type. In general, *DgIAA21* transgenic plants had lower drought tolerance than Col-0 plants.

## 3. Discussion

As one of the most outstanding gene families in mediated auxin signal transduction, the *Aux/IAA* family not only plays a key role in plant growth and development, but it also plays a significant role in responses to abiotic stresses [5,35]. The function of *Aux/IAA* genes has been deeply studied, mainly in model plants such as *Arabidopsis thaliana* and *Oryza sativa.* For *Arabidopsis thaliana*, *AtIAA8* interactions with *AtARF6* and *AtARF8* lead to abnormal flower organ development through a change in jasmonic acid level [36]. Mutants of *iaa16* display smaller rosettes and fewer lateral roots than the wild type [37]. For *Oryza sativa*, the overexpression of *OsIAA4* shows dwarfism and insensitivity to exogenous auxin [38]. Overexpressing *OsIAA31* reduces crown root formation and produces short leaf blades [39]. *OsIAA20* RNAi transgenic rice exhibits reduced drought and salt tolerance [32].

We identified 30 members of the *Aux/IAA* gene family in orchardgrass. The gene members of the *Aux/IAA* family exist with huge variation, from 1 to 119 members in different species, such as 1 in *Marchantia polymorpha* [40], 16 in *Solanum melongena* [41], 31 in *Oryza sativa* [42], 63 in *Glycine max* [43], and 119 in *Brassica napus* [14]. Gene duplication events could play a decisive role in functional diversity and evolutionary mechanisms [44]. In this study, four pairs of tandem duplications and seven pairs of segmental duplications were confirmed in orchardgrass. The rate of gene duplication was lower than that in Arabidopsis and rice [8,34,42].

Drought is one of the most serious threats for agriculture worldwide [45]. Increasing evidence has shown that *Aux/IAA* proteins can enhance drought stress response [5,46]. The overexpression of *OsIAA6* in rice enhanced drought stress tolerance through potentially regulating the auxin biosynthesis gene’s expression level [47]. In addition, some genes of the *Aux/IAA* family, including *MdIAA9* in apple and *OsIAA18* in rice, have also been shown to increase drought stress tolerance [33,48]. In this study, *DgIAA6* was clustered with *OsIAA18* and *OsIAA6. DgIAA6* transformed into INVScI yeast showed that *DgIAA6* can enhance drought resistance. However, *Aux/IAAs* also play a role as a negative regulator in response to drought stress. For example, the expression level of *IAA4* in *Vitis vinifera* was decreased under drought treatment [46]. Low cell recovery in yeast and a low gene expression level under drought stress showed that *DgIAA21* could play a role as a negative regulator in response to drought stress.

Previous studies have shown that photosynthetic machinery is remarkably impacted by drought stress and can especially cause a decrease in chlorophyll content [49,50]. Photosynthesis fuels lots of metabolic processes through triggering the process of conversion of light energy into chemical energy. The Pn and total chlorophyll content of *DgIAA21-*overexpressing plants was significantly decreased compared to Col-0 plants under control conditions, implying that the overexpressing plants may have less energy for defense when suffering abiotic stress. The chlorophyll fluorescence and total chlorophyll content in overexpressing lines were lower than in Col-0 lines under drought stress conditions, which can cause a reduction in PQ and PSII electronic transport [51]. In addition, the chlorophyll synthesis gene *CDR1′s* expression level was significantly decreased in *DgIAA21*-overexpressing plants under normal and drought conditions, suggesting that *DgIAA21* may negatively regulate chlorophyll biosynthesis to reduce drought stress.

It is important for plants to maintain a moisture balance to defend against drought stress [52,53]. Longer roots can enhance the ability of a plant obtain water from deeper soil compared with short roots to cope with drought conditions [54,55]. The relative water content and root length of transgenic Arabidopsis plants containing *DgIAA21* were lower than those of control plants under drought stress conditions. Moreover, *DgIAA21*-overexpressing plants exhibited poor WUE and higher water loss. These results indicated that *DgIAA21-*overexpressing plants were less able to maintain a moisture balance, which may be an important factor that decreases drought tolerance.

Drought can cause lipid peroxidation, membrane damage, oxidation, and even cell death through a rapid increase in ROS [56,57]. The antioxidant enzyme gene *APX1* and the Na+/H+ antiporter gene *NHX2 in DgIAA21*-overexpressing plants showed lower expression levels than Col-0, suggesting *DgIAA21* may negatively regulate drought responses via reducing the expression levels of the *APX1* and *NHX2* genes in Arabidopsis. As an indicator of ROS, MDA concentration reflects the degree of stress [58]. Our results showed that *DgIAA21*-overexpressing plants were more sensitive than Col-0 plants under drought stress conditions, as indicated by the increased level of MDA.

## 4. Materials and Methods

### 4.1. Analyzing Expression Patterns of Aux/IAAs in Orchardgrass

The expression patterns of *Aux/IAA* genes in root, stem, leaf, spike, and flower tissues were obtained from the orchardgrass reference genome database [59]. To analyze expression patterns under different stresses, the seeds of the ‘2006-1′ cultivar were germinated and grown in pots filled with quartz. Seedlings were irrigated using Hoagland’s solution for four weeks. Then, these plants were treated with exogenous auxin (100 μM IAA), drought stress (20% PEG 6000 (*w*/*v*)), salt stress (200 mM NaCl), high-temperature stress (37 °C) and low-temperature stress (4 °C), respectively. Samples of leaves were collected after 0 h, 2 h, 4 h, and 8 h under different stress treatments. Three biological replicates from each plant were harvested at each time point. All samples were placed into liquid nitrogen and stored at −80 °C. Total RNA was extracted using an RNA kit (RNA simply total RNA Kit, Tiangen, Beijing, China). cDNA was synthesized using a PrimeScript™ II 1st Strand cDNA Synthesis Kit (Takara, Dalian, China). Finally, gene expression was determined with qRT-PCR using an SYBR Primix Ex Taq kit (TaKaRa). Primers were designed using NCBI. The reference genes were the *β-actin* gene and *GADPH* gene of orchardgrass (Appendix A). The relative expression level of selected genes was calculated using the 2^−∆∆Ct^ calculation method [60].

### 4.2. Yeast Transformation and Verification of Drought Stress

To further analyze the functions in drought of *DgIAAs*, some genes of clade A (*DgIAA3*, *DgIAA14*, *DgIAA21*) and clade B (*DgIAA4*, *DgIAA6*) of orchardgrass were randomly selected to connect the yeast expression system. The open reading frames (ORFs) of these genes were inserted into the pYES2 vector. The specific primers for gene amplification are listed in Appendix A. Then, the recombinant plasmid and empty vector were transformed into the yeast strain INVScI (Weidi Biotechnology Co., Ltd., Shanghai, China). The correct single colony was selected and then incubated in SD-Ura liquid medium containing 2% (*w*/*v*) galactose at 28 °C and 200 rpm until cell densities reached an OD_600_ of 2.0. Yeast suspensions were diluted to 10^0^, 10^−1^, 10^−2^, 10^−3^, 10^−4^, and 10^−5^ with ddH_2_O. Finally, diluted yeast cells were spotted on yeast peptone glucose (YPG) plates containing 0, 1.5 mol/L, and 2.5 mol/L sorbitol for phenotypic analysis.

### 4.3. Germination and Root Length Assays

Sterilized seeds from T_3_ transgenic and Col-0 Arabidopsis plants were sown on 1/2 MS medium containing 0 mM, 200 mM, or 300 mM mannitol. The germination rate was computed 10 days after sowing. The seeds of T_3_ transgenic and Col-0 lines were sown on 1/2 MS medium and then transferred to 1/2 MS medium containing 0 mM, 200 mM, or 300 mM mannitol for 7 days to measure the root length of plants.

### 4.4. Drought Tolerance Analysis of Transgenic A. thaliana Plants

The seeds of Col-0 and homozygous T_3_ *DgIAA21*-overexpressing Arabidopsis were sown on soil and kept in a phytotron for 3 weeks to use for further treatments. In the drought treatment, water was withheld from plants for 14 days, and then watered for 4 days before plant recovery was recorded [61]. Normally watered Col-0 and *DgIAA21*-overexpressing Arabidopsis plants were considered the control. Malondialdehyde content was measured using the MDA Assay Kit (Grace Biotechnology Co., Ltd., Suzhou, China). The water loss, relative water content, relative electrolyte leak, and total chlorophyll content were determined using a previously described method [62]. Chlorophyll fluorescence was measured and recorded using a Pocket PEA Chlorophyll Fluorimeter (Hansatech, King’s Lynn, UK). The photosynthesis parameters, including intercellular CO_2_ concentration, stomatal conductance, transpiration rate, net photosynthetic rate, and water use efficiency, were measured via CIRAS-3 (Boston, MA, USA).

Three-week-old plants of the Col-0 and *DgIAA21*-overexpressing varieties were treated with 20% PEG 6000 (*w*/*v*), and the leaves were collected 0 h, 2 h, 4 h, 6 h, 12 h, and 24 h after treatment. Then, 13 stress-related genes and 8 chlorophyll-related gene were further analyzed via qRT-PCR [24,61]. The reference genes were the *Actin* gene and the *Actin2* gene of Arabidopsis. The specific primers are listed in Appendix A.

### 4.5. Subcellular Localization of DgIAA21 Protein

In order to explore the location of *DgIAA21*, the ORF of *DgIAA21* was cloned into a pAN580-35S-GFP vector. The specific primers are presented in Appendix A. The empty vector of pAN580-35S-GFP was used as the control. The recombined fusion vector and empty control vector were transferred into the rice protoplasts as described in [63]. Subcellular localizations were visualized using a Nikon C2-ER laser confocal microscope system with excitation at 488 nm and emission at 510 nm.

### 4.6. Plasmid Construction and Transformation of A. thaliana

The ORF of *DgIAA21* was inserted into a pCAMBIA1300-35S-GUS vector, and the specific primers are listed in Appendix A. The recombinant plasmid was introduced into *Agrobacterium tumefaciens* GV3101, which was transformed into Arabidopsis via the floral dip method [64]. The transgenic Arabidopsis plants were screened on a Petri dish with 1/2 Morishige and Skoog (MS) medium containing 30 ug/mL hygromycin. Transgenic plants were verified via qRT-PCR amplification with *DgIAA21*-qPCR-F/R primers (Appendix A). Homozygous lines in the T_3_ generation overexpressing *DgIAA21* were used in subsequent phenotypic and stress assays.

### 4.7. Identification and Protein Properties of the Aux/IAA Gene Family in Orchardgrass

The total protein sequences of orchardgrass were obtained from the latest version of the orchardgrass reference genome data [59]. The Aux/IAA protein sequences of *Arabidopsis thaliana* and *Oryza sativa* were downloaded from the NCBI gene database (https://www.ncbi.nlm.nih.gov, accessed on 1 August 2023) according to the Gene ID [34,65]. The *Arabidopsis thaliana* Aux/IAA protein sequences were used as queries to search the orchardgrass genome for putative Aux/IAA sequences, which were obtained based on a BLASTP search in NCBI. Further support for the putative Aux/IAA protein sequences was gained using hidden Markov model (HMM) profiles with a domain of *Aux/IAA* (PF02309). The protein length in amino acids, molecular weight, isoelectric point, instability index, aliphatic index, and grand average of hydropathicity of the *Aux/IAA* gene family were calculated using ProtParam (https://web.expasy.org/protparam/, accessed on 5 August 2023). The subcellular localization of proteins was predicted using WoLF PSORT (https://wolfpsort.hgc.jp/, accessed on 5 August 2023).

### 4.8. Phylogenetic and Synteny Analysis

The intron–exon structure and chromosome location of orchardgrass Aux/IAAs were analyzed using TBtools. MEME Suite was used to analyze the conserved motifs of orchardgrass Aux/IAA proteins (https://meme-suite.org/meme/tools/meme, accessed on 7 August 2023). A maximum likelihood (ML) phylogenetic tree was built using MEGA 7.0 software with 1000 bootstrap iterations [66]. The tree was further edited using the Interactive Tree of Life (iTOL) v5.7 web tool [67]. The collinear regions in orchardgrass *Aux/IAA* genes, as well as the collinear regions of *Aux/IAA*s among orchardgrass, Arabidopsis, and rice, were explored using MCScanX software https://mybiosoftware.com/tag/mcscanx (accessed on 5 November 2023) [68].

### 4.9. Statistical Analysis

The data of histogram was analyzed through an analysis of variance (ANOVA) using SPSS 22.0 software (IBM, Armonk, NY, USA), followed by a Tukey’s test (*p* < 0.05). The histogram was further produced using GraphPad Prism 9.2.0 software (GraphPad, Boston, MA, USA).

## 5. Conclusions

We identified 30 *Aux/IAAs* in orchardgrass, which were classified into two clades that showed divergence between dicots and monocots. Segmental duplication may have played a leading role during the evolution of the *Aux/IAA* gene family in orchardgrass. We isolated and identified the *DgIAA21* gene in orchardgrass. Heterologous expression of the *DgIAA21* gene reduced drought tolerance in yeast. The overexpression of *DgIAA21* in Arabidopsis had poor WUE and low total chlorophyll content and Pn in control conditions, which may have caused decreased tolerance when overexpressing plants suffered drought stress. Moreover, compared with Col-0 plants, higher levels of MDA content and relative electrolyte leakage and lower *APX1* and *NHX2* gene expression levels in transgenic plants were observed under drought stress, which also proved that *DgIAA21* transgenic plants were more sensitive than Col-0 plants. This study provides valuable information for the biological function of *DgAux/IAA* in the regulatory mechanisms of abiotic stress.

## Figures and Tables

**Figure 1 ijms-24-16184-f001:**
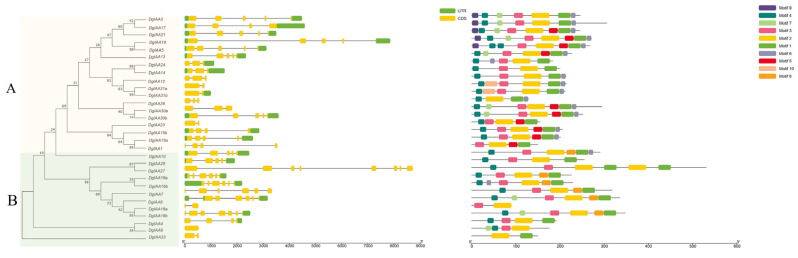
Gene cluster, structure, and conserved motifs analysis of *DgAux/IAA* gene family. The phylogenetic tree, intron–exon structure distribution and conserved motif distribution of *DgAux/IAAs* were sequenced from left to right. The *DgAux/IAAs* were divided into two clades: the yellow presents clade (**A**) and the green presents clade (**B**) in the phylogenetic tree section. The yellow boxes indicate exons, and the gray lines indicate introns in the intron–exon structure section. Ten putative motifs are shown in different colored boxes in the conserved motif analysis section.

**Figure 2 ijms-24-16184-f002:**
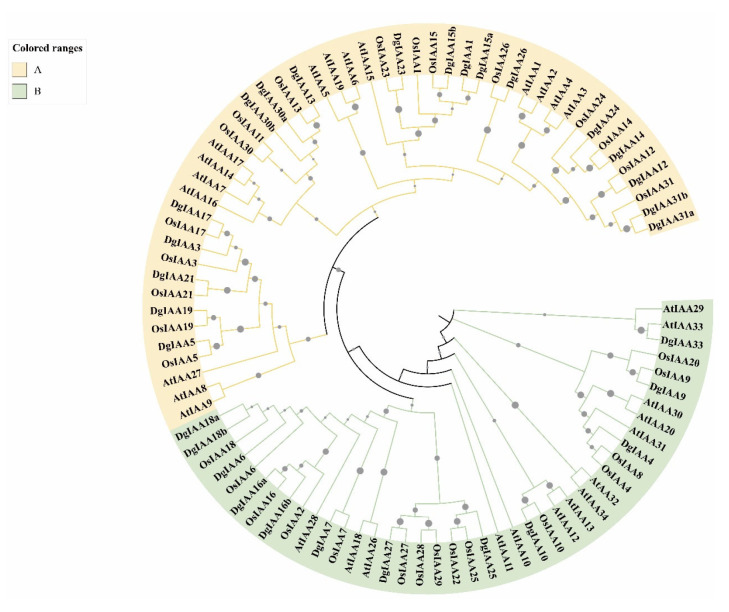
Construction of phylogenetic trees of *Aux/IAA* gene family among *DgIAAs*, *AtIAAs*, and *OsIAAs*. Yellow regions represent clade A, while the green areas indicate clade B.

**Figure 3 ijms-24-16184-f003:**
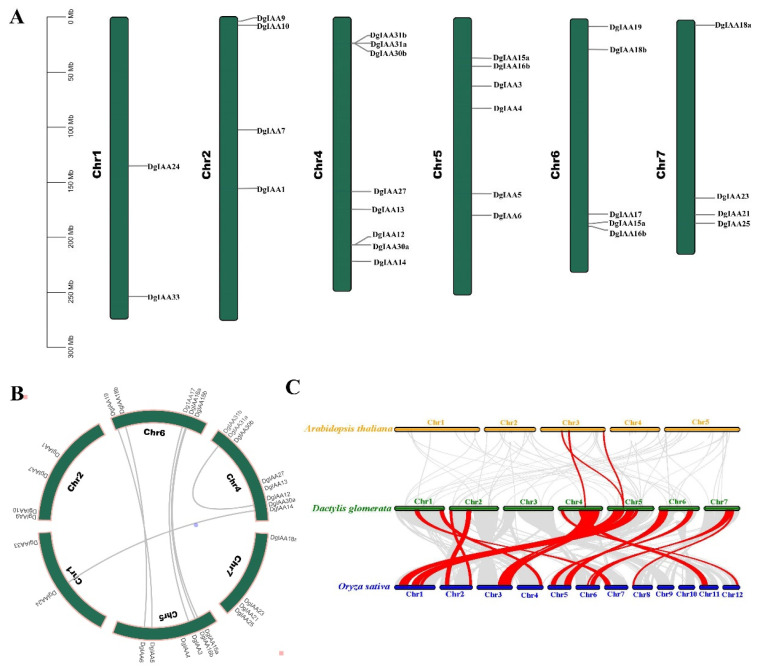
Chromosome location and syntenic analysis of *Aux/IAA* gene family. (**A**) Distribution of *DgAux/IAAs* on orchardgrass chromosomes. (**B**) Segmental duplication of *DgAux/IAA* genes. Different gray lines represent different gene pairs on *Aux/IAA* family of orchardgrass. (**C**) Syntenic analysis of *Dactylis glomerata*, *Arabidopsis thaliana*, and *Oryza sativa Aux/IAA* genes. Highlighted red lines show the syntenic relationships among *Dactylis glomerata*, *Arabidopsis thaliana*, and *Oryza sativa Aux/IAA* genes.

**Figure 4 ijms-24-16184-f004:**
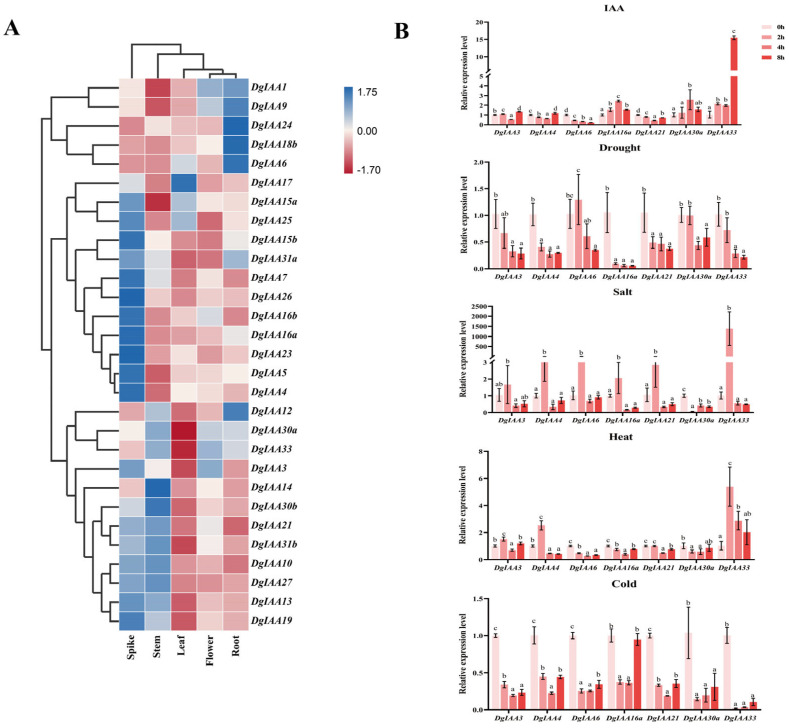
The expression pattern of *DgAux/IAA* genes. (**A**) The expression profiles of *Aux/IAA* genes in different tissues. (**B**) The expression analysis of seven genes of *Aux/IAA* under IAA, drought, salt, heat, and cold stresses. Bars represent mean standard deviation (SD). The different letters above the bars indicate significant differences among different times (*p* < 0.05, according to Tukey’s multiple range test).

**Figure 5 ijms-24-16184-f005:**
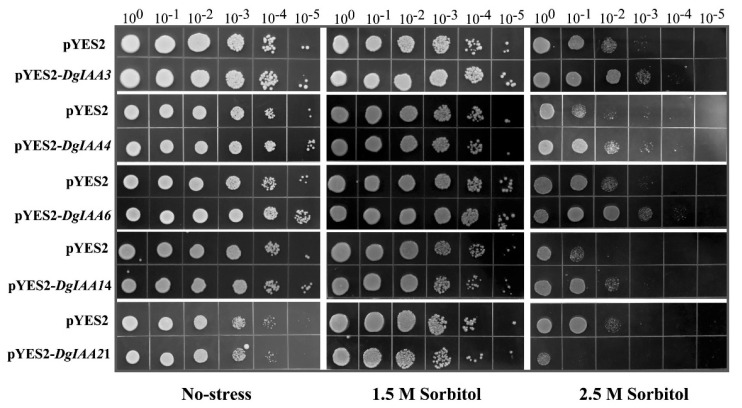
Phenotypic growth assays of pYES2-*DgIAA3*, -*DgIAA4*, -*DgIAA6*, -*DgIAA14*, -*DgIAA21,* and empty vector under drought stress. INVScI yeast cells were dotted on YPG medium in 2 uL aliquots and were diluted 10 times in turn using ddH_2_O.

**Figure 6 ijms-24-16184-f006:**
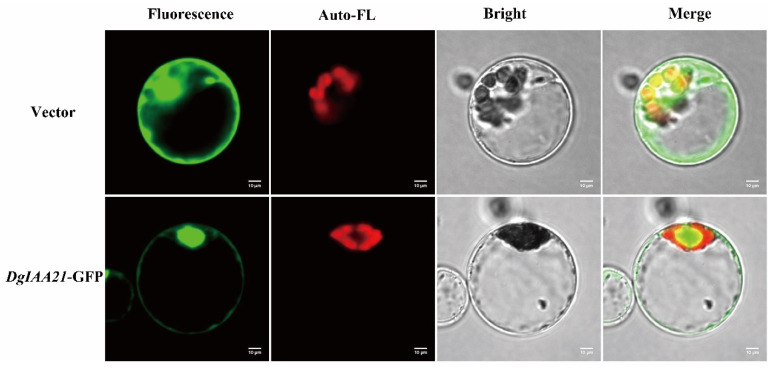
Subcellular localization of *DgIAA21* in rice protoplasts. The green florescent protein signals obtained via confocal microscopy showed that fusion protein *DgIAA21*-GFP were localized primarily in the nucleus, and minority in nucleus and cytoplasm. The green florescent protein fields, chlorophyll auto fluorescent fields, bright fields, and merge fields are indicated in images from left to right. Scale bar = 10 um.

**Figure 7 ijms-24-16184-f007:**
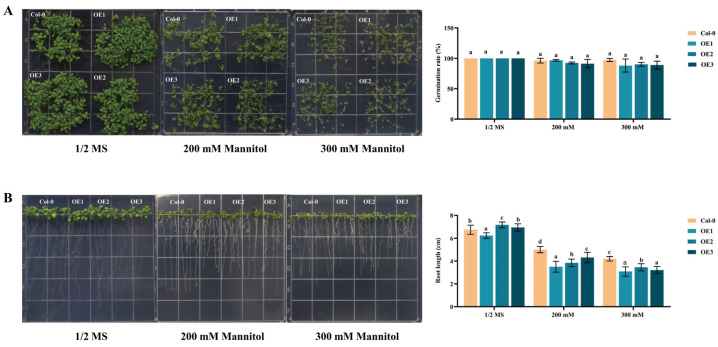
Effects of drought stress on germination and root length of Col-0 and overexpression *DgIAA21* transgenic Arabidopsis lines. (**A**) Phenotypes and germination rates of Col-0 and *DgIAA21* transgenic lines subjected to 1/2 MS, 200 mM Mannitol, and 300 mM Mannitol treatments. (**B**) Phenotypes and root lengths in *DgIAA21* transgenic lines and Col-0 lines under same treatment conditions. Bars represent mean standard deviation (SD). The different letters above the bars indicate significant differences between Col-0 and three transgenic lines (*p* < 0.05, according to Tukey’s multiple range test).

**Figure 8 ijms-24-16184-f008:**
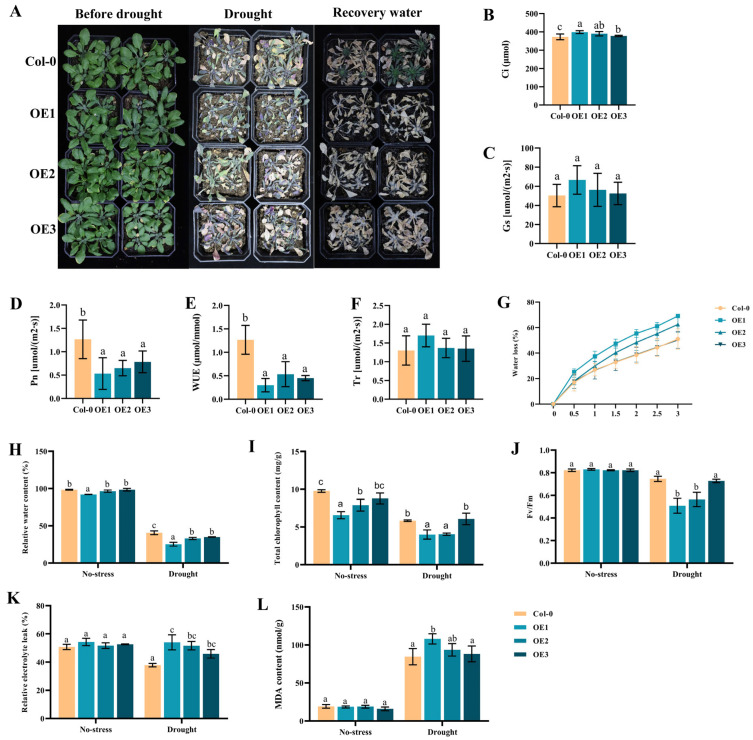
Overexpression of *DgIAA21* transgenic *Arabidopsis* plants were sensitive to drought tolerance. Three-week-old *A.thaliana* plants with Col-0 and *DgIAA21*- overexpression were used to drought experiment. (**A**) Appearance of plants were given drought tolerance through withholding watering for 14 days and recovery watering for 4 days. Comparison of photosynthesis parameters and several physiological parameters in Col-0 and transgenic plants: (**B**) intercellular CO_2_ concentration, (**C**) stomatal conductance, (**D**) net photosynthetic rate, (**E**) water use efficiency, (**F**) transpiration rate, (**G**) water loss, (**H**) relative water content, (**I**) total chlorophyll content, (**J**) chlorophyll fluorescence, (**K**) relative electrolyte leakage, and (**L**) MDA content. (**B**–**G**) were measured under normal condition, while (**H**–**L**) were detected under normal and drought conditions. Bars represent mean standard deviation (SD). The different letters above the bars indicate significant differences between Col-0 and three transgenic lines (*p* < 0.05, according to Tukey’s multiple range test).

**Figure 9 ijms-24-16184-f009:**
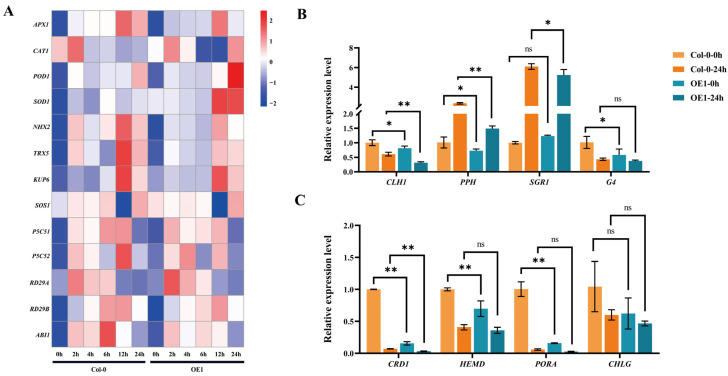
The expression of stress-related genes and chlorophyll-related genes of Col-0 and *DgIAA21* OE1 line under 20% PEG 6000 treatment. (**A**) The heatmap of expression levels of 13 stress-related genes after 0 h, 2 h, 4 h, 6 h, 12 h, and 24 h under drought stress. (**B**) The changes in the expression level of chlorophyll degradation pathway genes (**C**) and chlorophyll synthesis pathway genes (**C**) after 0 h and 24 h under drought stress. Error bars represent mean standard deviation (SD). Asterisks (*) represent significant differences compared with the Col-0 plants at the same time. *, *p* < 0.05; **, *p* < 0.01, ns: non-significant.

## Data Availability

Data are contained within the article and Appendix A.

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
