# Peer review of "Genome-Wide Characterization of the *Aux/IAA* Gene Family in Orchardgrass and a Functional Analysis of *DgIAA21* in Responding to Drought Stress"

_ijms, 2023, doi:10.3390/ijms242216184_

Round 1
Reviewer 1 Report
Comments and Suggestions for Authors
The article is scientifically sound and can be accepted after minor changes.
The manuscript titled 'Genome-Wide Characterization of the Aux/IAA Gene Family in Orchardgrass and Role of the Negative Regulator DgIAA21 in Drought Stress' has been written well and offers some novel findings that could be explored further in future studies. However, some comments should be addressed. 1. In line 12, the authors mentioned that Orchardgrass has a high level of drought tolerance. If this is the case, what was the actual reason for identifying the drought tolerance genes? 2. What is the mechanism by which genes DgIAA3, DgIAA4, DgIAA6, and DgIAA14 regulate drought tolerance? Do these genes encode any proteins or enzymes? 3. The abstract could be improved by clearly indicating the core findings of the study. 4. The keywords should be different from the title. 5. The introduction could be rearranged to start with the significance of Orchardgrass, followed by abiotic stresses, drought stress, and so on. 6. The quality of Figure 1 is not good. 7. Figure 2, the "phylogenetic trees," shows many genes. Have the authors verified these genes? How many of them are real and functionally validated? 8. Although the results are presented in a detailed way, the quality of Figure 4 is not good. 9. In line 142, the authors mentioned that genes were expressed in spike, stem, leaf, flower, and root tissues, but in the introduction section, it was mentioned that drought tolerance was based on physio-biochemical traits. An increase in root length, shoot length, or leaf area is not a key indicator of drought tolerance. The most important factor is the activation of antioxidant enzymes. 10. Figure 8 should be of good quality and should be attractive and readable. 11. The role of "IAA" should be mentioned in the discussion section. 12. The discussion needs more detail, and the conclusion is missing. 13. The English language needs minor improvement. 14. Carefully check the formatting of text and references.
Comments on the Quality of English LanguageMinor English editing is required.
Author Response
|
Response to Reviewer 1 Comments
|
||
|
1. Summary |
|
|
|
Thank you very much for taking the time to review this manuscript. Please find the detailed responses below and the corresponding revisions/corrections highlighted/in track changes in the re-submitted files. |
||
|
2. Questions for General Evaluation |
Reviewer’s Evaluation |
Response and Revisions |
|
Is the work a significant contribution to the field? |
|
[Please give your response if necessary. Or you can also give your corresponding response in the point-by-point response letter. The same as below] |
|
Is the work well organized and comprehensively described? |
|
|
|
Is the work scientifically sound and not misleading? |
|
|
|
Are there appropriate and adequate references to related and previous work? |
|
|
|
Is the English used correct and readable? |
|
|
|
3. Point-by-point response to Comments and Suggestions for Authors |
|
|
|
Comments 1: In line 12, the authors mentioned that Orchardgrass has a high level of drought tolerance. If this is the case, what was the actual reason for identifying the drought tolerance genes? |
||
|
Response 1: Thank you for pointing this out. The reason for identifying the drought tolerance genes can help to provide potential candidate genes for grass plant breeding of drought stress. |
||
|
Comments 2: What is the mechanism by which genes DgIAA3, DgIAA4, DgIAA6, and DgIAA14 regulate drought tolerance? Do these genes encode any proteins or enzymes? |
||
|
Response 2: Thank you for pointing this out. I am sorry that I did not further analyze the molecular mechanism of DgIAA3, DgIAA4, DgIAA6, and DgIAA14 in this paper because we need to mainly analyze the molecular mechanism of DgIAA21 in drought stess, and we will continue to explore molecular research on DgIAA3, DgIAA4, DgIAA6, and DgIAA14 genes in drought tolerance in the following studies. Previous study showed that the OsIAA18 protein, which encoded by OsIAA18 gene, was localized in the nucleus and was a transcription activator (Li et al. 2020). Thus, the gene of DgIAA3, DgIAA4, DgIAA6, or DgIAA14 could be transcribed and translated into certain proteins. But we only studied the structural characteristics and potential functions of these genes and did not conduct further studies on the properties of the proteins they encode in this study. |
||
|
Comments 3: The abstract could be improved by clearly indicating the core findings of the study. |
||
|
Response 3: Thank you for pointing this out. We agree with this comment. Therefore, we have modified the part of abstract in line 10-24. |
||
|
Comments 4: The keywords should be different from the title. |
||
|
Response 4: Thank you for pointing this out. We agree with this comment. Therefore, we have modified the keywords in line 25. |
||
|
Comments 5: The introduction could be rearranged to start with the significance of Orchardgrass, followed by abiotic stresses, drought stress, and so on. |
||
|
Response 5: Thank you for pointing this out. We rearranged the introduction to start with the significance of orchardgrass, followed by the abiotic stresses, drought stress. The modification of introduction is shown on line 28-116. |
||
|
Comments 6: The quality of Figure 1 is not good. |
||
|
Response 6: We feel sorry for the inconvenience brought to the reviewer. I am sorry for that the figures we provided in manuscript did not have enough clarity. So, we will re-provide all figures in JPEG format. |
||
|
Comments 7: Figure 2, the "phylogenetic trees," shows many genes. Have the authors verified these genes? How many of them are real and functionally validated? |
||
|
Response 7: Thank you for pointing this out. The genes in phylogenetic trees including Aux/IAA genes in Dactylis glomerata, Arabidopsis thaliana, and Oryza sativa species. The Aux/IAA genes in Arabidopsis thaliana, and Oryza sativa were verified in provide studies(Jain et al. 2006; Liscum and Reed 2002). We predicted the putative genes for Aux/IAA genes using bioinformatics analysis in this study. In addition, we also explored the potential function of Aux/IAA gene family of orchardgrass in abiotic stresses. |
||
|
Comments 8: Although the results are presented in a detailed way, the quality of Figure 4 is not good. |
||
|
Response 8: We feel sorry for the inconvenience brought to the reviewer. I am sorry for that the figures we provided in manuscript did not have enough clarity. So, we will re-provide all figures in JPEG format. |
||
|
Comments 9: In line 142, the authors mentioned that genes were expressed in spike, stem, leaf, flower, and root tissues, but in the introduction section, it was mentioned that drought tolerance was based on physio-biochemical traits. An increase in root length, shoot length, or leaf area is not a key indicator of drought tolerance. The most important factor is the activation of antioxidant enzymes. |
||
|
Response 9: Thank you for pointing this out. We mentioned the expression of DgIAAs in spike, stem, leaf, flower, and root tissues mainly to understand the characteristic in different tissues for this gene family rather than showed these traits were key indicator in drought tolerance. |
||
|
Comments 10: Figure 8 should be of good quality and should be attractive and readable. |
||
|
Response 10: We feel sorry for the inconvenience brought to the reviewer. I am sorry for that the figures we provided in manuscript did not have enough clarity. So, we will re-provide all figures in JPEG format. |
||
|
Comments 11: The role of "IAA" should be mentioned in the discussion section. |
||
|
Response 11: Thank you for your suggestions. We agree with this comment. Therefore, we added the information related to the role of "IAA" in the discussion section. The corrections are listed in line 364-371. |
||
|
Comments 12: The discussion needs more detail, and the conclusion is missing. |
||
|
Response 12: Thank you for your suggestions. We agree with this comment. Therefore, we added more details in discussion. For conclusion, we put the part of “conclusion” after the part of “Materials and Methods” according to the layout in Int. J. Mol. Sci., which can be found on line 667-679. |
||
|
Comments 13: The English language needs minor improvement. |
||
|
Response 13: Thank you for your suggestions. We agree with this comment. Therefore, we have tried our best to polish the language in the revised manuscript. |
||
|
Comments 14: Carefully check the formatting of text and references. |
||
|
Response 14: Thank you for pointing this out. We agree with this comment. Therefore, we have carefully checked the formation of text and reference. |
||
|
4. Response to Comments on the Quality of English Language |
||
|
Point 1: Minor English editing is required |
||
|
Response 1: Thank you for your suggestions. We agree with this comment. Therefore, we have tried our best to polish the language in the revised manuscript. |
||
|
5. Additional clarifications |
||
|
|
||

Reviewer 2 Report
Comments and Suggestions for Authors
Dear Authors,
You identified 30 probable genes from Aux/IAA family in Orchardgrass (Dactylis glomerata). Some other complementary experiments and analysis are done leading you to find out an interesting output about the negative role for DgiAA21 under drought stress. I believe your study can be greatly appreciated for those scientists interested in forage grass. The results can also draw more attention to the effect of drought stresses in Orchardgrass. Your methodology seems acceptable; however, I think the paper can be improved after covering/addressing some drawbacks.
Here are my main concerns about the manuscript.
- In the phylogenetic analysis of the Orchardgrass protein sequences together with Arabidopsis and Rice, I disagree with your conclusion out of tree. I don’t see clustering in Orchardgrass and Arabidopsis (as you mentioned both are dicots) (fig.2). The separation of these two with Rice (as monocot) again is not as clear as you stated in manuscript. It seems in this part of analysis they clustered more or less random, or at least not the way you described. I also didn’t understand how you split the tree into two clades (A and B) (fig 1); the tree doesn’t support that well.
- I couldn’t find how you decided to select some certain genes out of 30 for expression analysis, and why you select only DgIAA21 for over expression assay. These are confusing.
- The abstract is not prepared well, you need to rewrite that part.
- I also recommend considering editing/changing the title.
You may also find the file enclosed containing detailed corrections and suggestions. I left some corrections/suggestions regarding the above-mentioned concerns and some others.
I hope these suggestions can have a minor positive impact on improving your valuable work.
Cheers,
Reviewer

--
Reviewer 3 Report
Comments and Suggestions for Authors
In the Abstract the use of species common name is fine, but it should be put also the latin name of the species used.
In the Abstract the results reached are too generic.
In the introduction I would suggest to insert the reference Int. J. Mol. Sci. 22, 6378. https://doi.org/10.3390/ijms22126378
Please renumber all the Tables. It is a bit strange that the first table mentioned is the Table S3.
Delete "different" from rows 89 and 114.
Fig 1 is unreadable as also others. Moreover, it is divided into 3 section, but there is not description in the Caps.
rows 99-101 If you leave Fig 2 between brackets the sentence seems to be uncompleted. But in any case please rephrase it.
Fig 2 what does "colored range" mean? Moreover, there is a wrong interpretation it is not true that all the green belong to the same clade.
On Row 121 DgIAA31b and DgIAA30b; and DgIAA31a and DgIAA30b. Actually, seems all 3 close. Please rewrite.
In Fig 9 caps it is not properly describe what star and bars (horizontal) indicate. i.e. which comparisons were considered.
Row 249-250 It is well known that number of genes among species is almost the same despite of a great genome size differences. Moreover sentence in rows 250-251 partially contradict the above one.
Row 255-256 Probably it is not so strange since both are cereals??? So rather than that probably would be better the results confirm that also for ###
Row 282 delete "in plants".
Row 282-283 please see also Agronomy, 12(6), 1329. doi:10.3390/agronomy12061329
In M&M I would suggest to change the order of section into:
4.3; 4.4; 4.7; 4.8; 4.5; 4.6; 4.1; 4.2; 4.9
In any case if not differently justify 4.1 and 4.2 seems to be better putting just before 4.9
4.9 should be better described. Which were the comparison in the ANOVA?
In 4.2 should be specify of was chose the DgIAA3, DgIAA4, DgIAA6, DgIAA14, and 336 DgIAA21
Comments on the Quality of English Language
.English is fine
